# Benchmarking uncertainty quantification for protein engineering

**Kevin P. Greenman**[1,2,3‡], **Ava P. Amini**[4*], **Kevin K. Yang**[4*]

**1** Department of Chemical Engineering, Catholic Institute of Technology, Cambridge, Massachusetts, United States of America, **2** Department of Chemistry, Catholic Institute of Technology, Cambridge, Massachusetts, United States of America, **3** Department of Chemical Engineering, Massachusetts Institute of Technology, Cambridge, Massachusetts, United States of America, **4** Microsoft Research, Cambridge, Massachusetts, United States of America

‡Work done in part during an internship at Microsoft Research
* ava.amini@microsoft.com (APA); yang.kevin@microsoft.com (KKY)

**Data Availability Statement:** The code for the models, uncertainty methods, and evaluation metrics in this work is available at https://github.com/microsoft/protein-uq and archived at https://zenodo.org/doi/10.5281/zenodo.7839141.

**Funding:** K.P.G. was supported by a Microsoft Research (https://www.microsoft.com/en-us/

## Abstract

Machine learning sequence-function models for proteins could enable significant advances in protein engineering, especially when paired with state-of-the-art methods to select new sequences for property optimization and/or model improvement. Such methods (Bayesian optimization and active learning) require calibrated estimations of model uncertainty. While studies have benchmarked a variety of deep learning uncertainty quantification (UQ) methods on standard and molecular machine-learning datasets, it is not clear if these results extend to protein datasets. In this work, we implemented a panel of deep learning UQ methods on regression tasks from the Fitness Landscape Inference for Proteins (FLIP) benchmark. We compared results across different degrees of distributional shift using metrics that assess each UQ method's accuracy, calibration, coverage, width, and rank correlation. Additionally, we compared these metrics using one-hot encoding and pretrained language model representations, and we tested the UQ methods in retrospective active learning and Bayesian optimization settings. Our results indicate that there is no single best UQ method across all datasets, splits, and metrics, and that uncertainty-based sampling is often unable to outperform greedy sampling in Bayesian optimization. These benchmarks enable us to provide recommendations for more effective design of biological sequences using machine learning.

## Author summary

Protein engineering has previously benefited from the use of machine learning models to guide the choice of new experiments. In many cases, the goal of conducting new experiments is optimizing for a property or improving the machine learning model. Many standard methods for these two tasks require good estimates of the uncertainty in the model's predictions. Several methods for quantifying this uncertainty exist and have been benchmarked on datasets from other domains (e.g. small molecules), but it is not clear whether these results also apply for proteins. To address this, we evaluated a range of uncertainty

research/) micro-internship and by the National Science Foundation Graduate Research Fellowship Program under Grant No. 1745302. The funders had no role in study design, data collection and analysis, decision to publish, or preparation of the manuscript.

**Competing interests:** The authors have declared that no competing interests exist.

quantification approaches on tasks derived from a protein-focused benchmark dataset. We tested performance on different degrees of distributional shift between the training and testing sets and on different representations of the sequences, and we assessed performance in terms of several standard metrics. Finally, we used the uncertainties for property optimization and model improvement. Our findings indicate that no single uncertainty estimation method excels across all scenarios. Moreover, uncertainty-based strategies for property optimization often did not outperform simpler methods that did not consider uncertainty. This research offers insights for the more efficacious application of machine learning in the realm of biological sequence design.

## Introduction

Machine learning (ML) has already begun to accelerate the field of protein engineering by providing low-cost predictions of phenomena that require time- and resource-intensive labeling by experiments or physics-based simulations [1]. It is often necessary to have an estimate of model uncertainty in addition to the property prediction, as the performance of an ML model can be highly dependent on the domain shift between its training and testing data [2]. Because protein engineering data is often collected in a manner that violates the independent and identically distributed (i.i.d.) assumptions of many ML approaches [3], tailored ML methods are required to guide the selection of new experiments from a protein landscape. Uncertainty quantification (UQ) can inform the selection of experiments in order to improve a ML model or optimize protein function through active learning (AL) or Bayesian optimization (BO).

In chemistry and materials science, several studies have benchmarked common UQ methods against one another on standard datasets and have used or developed appropriate metrics to quantify the quality of these uncertainty estimates [4–9]. These works have illustrated that the best choice of UQ method can depend on the dataset and other considerations such as representation and scaling. While some protein engineering work has leveraged uncertainty estimates, these studies have been mostly limited to single UQ methods such as convolutional neural network (CNN) ensembles [10] or Gaussian processes (GPs) [11, 12].

Gruver et al. compared CNN ensembles to GPs (using traditional representations and pretrained BERT [13] language model embeddings) in Bayesian optimization tasks [14]. They found that CNN ensembles are often more robust to distribution shift than other types of models. Additionally, they report that most model types have more poorly calibrated uncertainties on out-of-domain samples. However, a more comprehensive study of CNN UQ methods, evaluated using a variety of uncertainty quality metrics, has not been done. A comparison of uncertainty methods on different protein representations (e.g., one-hot encodings or embeddings from protein language models) in an active learning setting is also lacking.

In this work, we evaluate a panel of UQ methods for protein sequence-function prediction on a set of standardized, public protein datasets (Fig 1). Our chosen datasets included splits with varied degrees of domain extrapolation, which enabled method evaluation in a setting similar to what might be experienced while collecting new experimental data for protein engineering. We assessed each model using a variety of metrics that captured different aspects of desired performance, including accuracy, calibration, coverage, width, and rank correlation. Additionally, we compared the performance of the UQ methods on one-hot encoded sequence representations and on embeddings computed from the ESM-1b protein masked language model [15]. We find that the quality of UQ estimates are dependent on the landscape, task, and embedding, and that no single method consistently outperforms all others. We also

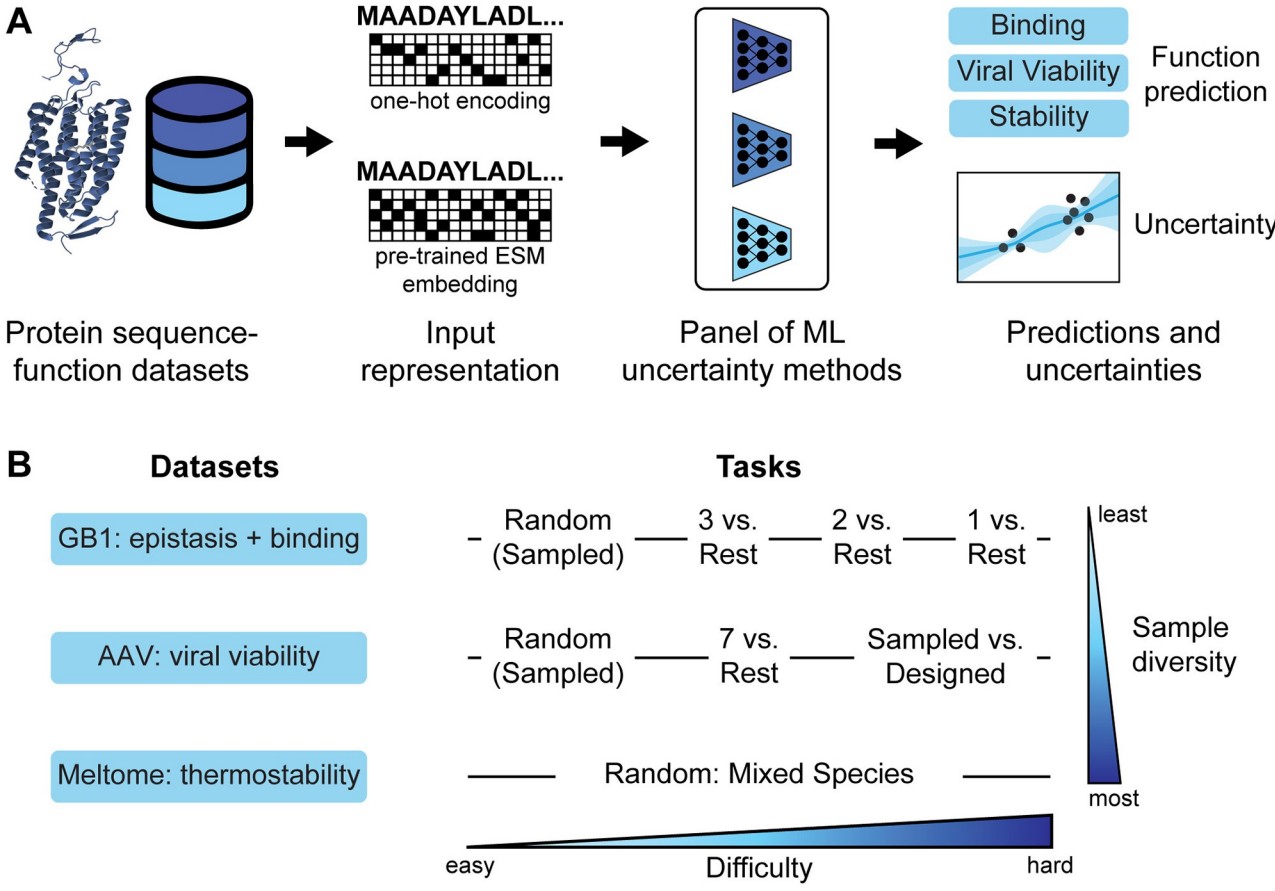

**Fig 1. Approach, datasets, and tasks.** (A) Schematic of the approach for benchmarking uncertainty quantification (UQ) in machine learning for protein engineering. A panel of UQ methods were evaluated on protein fitness datasets to assess the quality of the uncertainty estimates and their utility in active learning and Bayesian optimization. (B) Our study utilized three protein datasets/landscapes and different train-validation-test split tasks within each dataset. These datasets and tasks covered a range of sample diversities and domain shifts (task difficulties).

evaluated the UQ methods in an active learning setting with several acquisition functions, and demonstrated that uncertainty-based sampling often outperforms random sampling (especially in later stages of active learning), although better calibrated uncertainty does not necessarily equate to better active learning. Finally, we tested the UQ methods in Bayesian optimization and found that while BO typically outperformed random sampling, none were better than a greedy baseline. We envision that the understanding gained from this work will enable more effective development and application of UQ techniques to machine learning in protein engineering.

## Results and discussion

### Uncertainty quantification

Our first goal was to evaluate the calibration and quality of a variety of UQ methods. We implemented seven uncertainty methods for this benchmark: linear Bayesian ridge regression (BRR) [16, 17], Gaussian processes (GPs) [18], and five methods using variations on a convolutional neural network (CNN) architecture. The CNN implementation from FLIP [3] provided the core architecture used by our dropout [19], ensemble [20], evidential [21], mean-

variance estimation (MVE) [22], and last-layer stochastic variational inference (SVI) [23] methods. Additional model details are provided in the Methods section.

The landscapes used in this work were taken from the Fitness Landscape Inference for Proteins (FLIP) benchmark [3]. These include the binding domain of an immunoglobulin binding protein (GB1), adeno-associated virus stability (AAV), and thermostability (Meltome) data landscapes, which cover a large sequence space and a broad range of protein families. The FLIP benchmark includes several train-test splits, or tasks, for each landscape. Most of these tasks are designed to mimic common, real-world data collection scenarios and are thus a more realistic assessment of generalization than random train-test splits. However, random splits are also included as a point of reference. We chose 8 of the 15 FLIP tasks to benchmark the panel of uncertainty methods. We selected these tasks to be representative of several regimes of domain shift—random sampling with no domain shift (AAV/Random, Meltome/Random, and GB1/Random); the highest (and most relevant) domain-shift regimes (AAV/Random vs. Designed and GB1/1 vs. Rest); and less aggressive domain shifts (AAV/7 vs. Rest, GB1/2 vs. Rest, and GB1/3 vs. Rest). The Datasets section of the Methods provides notes on the nomenclature used for these tasks.

We trained the seven models on each of the eight tasks described above and evaluated their performance on the test set using the metrics described in the Evaluation Metrics section. We compare model calibration and accuracy in Fig 2 and the percent coverage versus average width relative to range in Fig 3. These figures illustrate the results for models trained on the embeddings from a pretrained ESM language model [15]; the corresponding results using one-hot encodings are shown in Figs A and B in S1 Appendix.

As expected, the splits with the least required domain extrapolation tend to have more accurate models (lower RMSE; Fig 2). However, the relationship between miscalibration area and extrapolation is less clear; some models are highly calibrated on the most difficult (highest domain shift) splits, while others are poorly calibrated even on random splits. There is no single method that performs consistently well across splits and landscapes, but some trends can be observed. For example, ensembling is often one of the highest accuracy CNN models, but also one of the most poorly calibrated. Additionally, GP and BRR models are often better calibrated than CNN models. For the AAV and GB1 landscapes (Fig 2a and 2c), model miscalibration area usually increases slightly while RMSE increases more substantially with increasing domain shift.

In addition to accuracy and calibration, we assessed each method in terms of the coverage and width of its uncertainty estimates. A good uncertainty method results in high coverage (ideally, the true value falls within the 95% confidence region 95% of the time) while still maintaining a small average width. The latter is necessary because predicting a very large and uniform value of uncertainty for every point would result in good coverage, so coverage alone is not sufficient. Fig 3 illustrates that many methods perform relatively well in either coverage or width (corresponding to the the top and left limits of the plot, respectively), but few methods perform well in both. Similarly to Fig 2, there is some observable trend that more challenging splits are further from the optimal part (upper left) of the plot; this trend is more clear for the GB1 splits (Fig 3b) than for the AAV splits. Most models trained on the AAV landscape (Fig 3a) have a similar average width/range ratio for all splits, but for the GB1 landscape (Fig 3c), this ratio typically increases as the domain shift increases. The locations of the sets of points for each model type shared some similarities across landscapes. CNN SVI often has low coverage and low width, CNN MVE often has moderate coverage and moderate width, and CNN Evidential and BRR often have high coverage and high width. These trends across landscapes could point to a general problem of under- or over-confidence with some model types, and indicates that post-hoc calibration may be necessary. The results for all prediction and

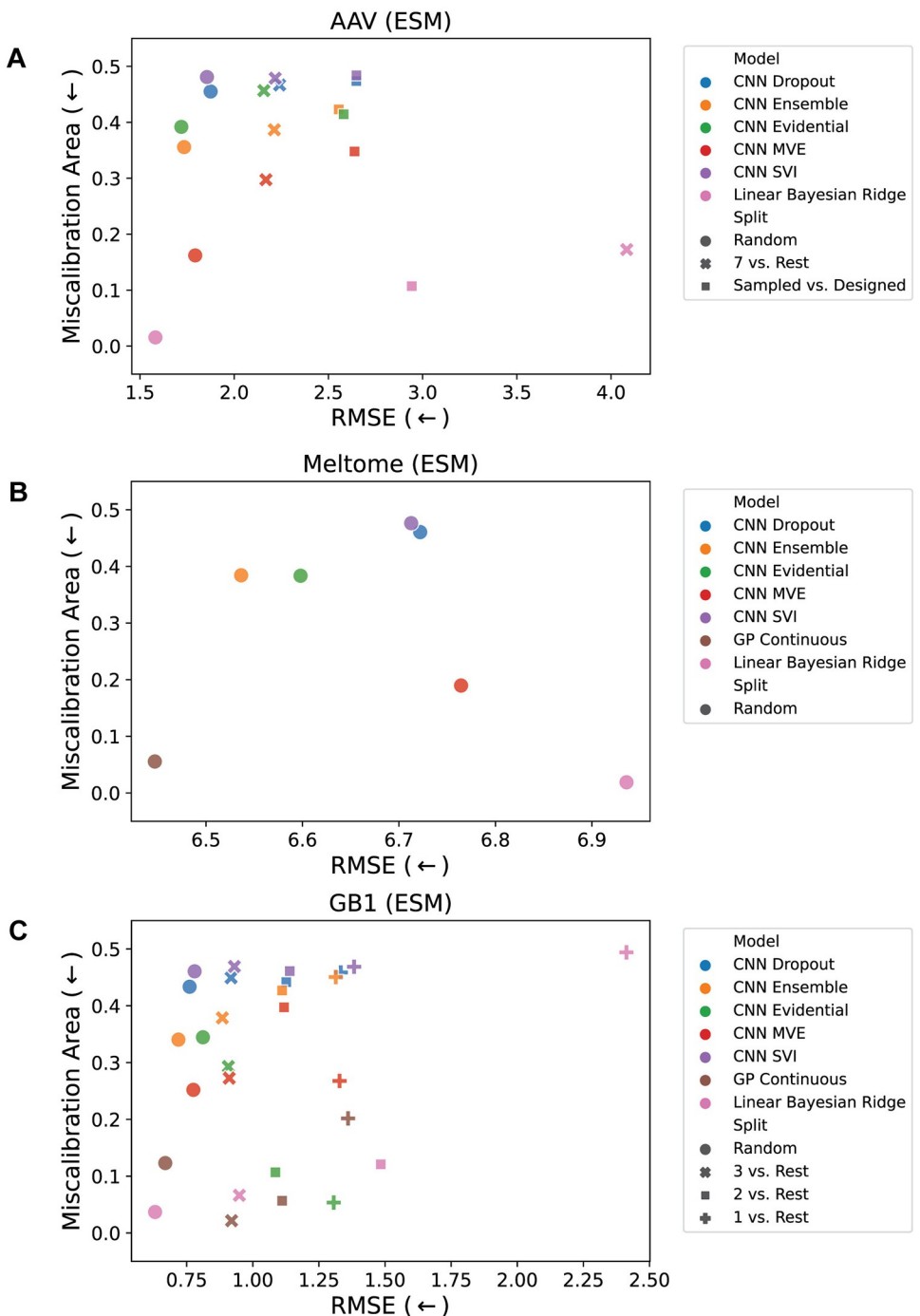

**Fig 2. Miscalibration area vs. root mean square error (RMSE).** For the (A) AAV, (B) Meltome, and (C) GB1 landscapes. Miscalibration area (also called the area under the calibration error curve or AUCE) quantifies the absolute difference between the calibration plot and perfect calibration. It is desirable to have a model that is both accurate and well-calibrated, so the best performing points are those closest to the lower left corner of the plots. Each point represents an average of 5 models trained using different random seeds for initialization of the CNN parameters and batching / stochastic gradient descent. Fig A in S1 Appendix shows the corresponding results for the OHE representation. See the Uncertainty Methods section for an explanation of points for which experiments were not feasible (e.g. there is no GP Continuous model result for the AAV landscape due to memory constraints for training these models).

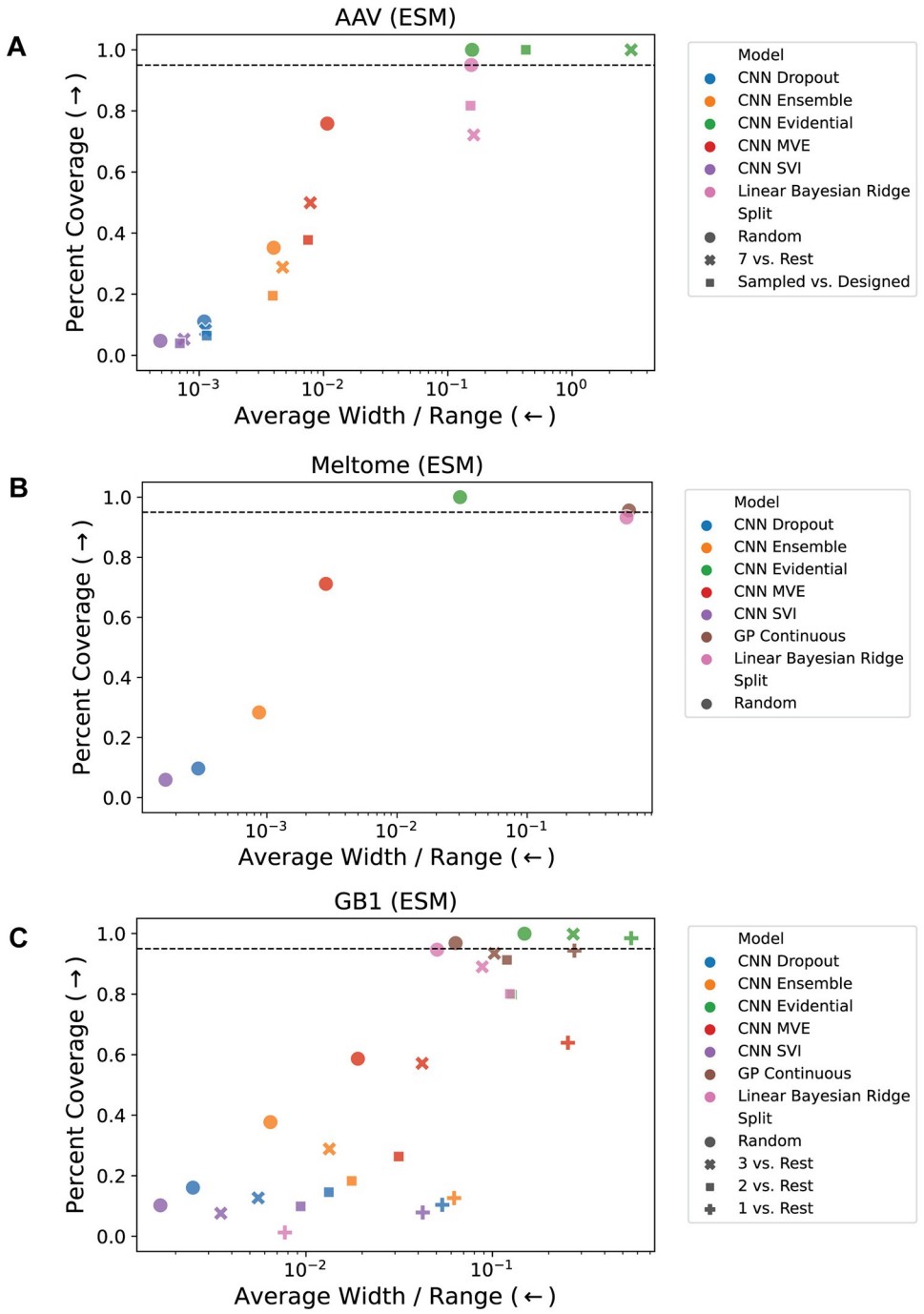

**Fig 3. Coverage vs. average width / range.** For the (A) AAV, (B) Meltome, and (C) GB1 landscapes. Coverage is the percentage of true values that fall within the 95% confidence interval ($\pm 2\sigma$) of each prediction, and the width is the size of the 95% confidence region relative to the range of the training set ($4\sigma/R$ where $R$ is the range of the training set). A good model exhibits high coverage and low width, which corresponds to the upper left of each plot. The horizontal dashed line indicates 95% coverage. Each point represents an average of 5 models trained using different random seeds for initialization of the CNN parameters and batching / stochastic gradient descent. Fig B in S1 Appendix shows the corresponding results for the OHE representation. See the Uncertainty Methods section for an explanation of several points for which experiments were not feasible (e.g. there is no GP Continuous model result for the AAV landscape due to memory constraints for training these models).

uncertainty metrics (along with their standard deviations across 5 different initialization seeds) are shown in Tables A to AR in S1 Appendix.

We next assessed how target predictions and uncertainty estimates depended on the degree of domain shift. Across datasets and splits, we compared the ranking performance of each method in terms of predictions relative to true values and uncertainty estimates relative to true errors (ESM in Fig 4 and one-hot encodings (OHE) in Fig C in S1 Appendix). The splits are ordered according to domain shift within their respective landscapes (lowest to highest shift from left to right). We observe that the rank correlation of the predictions to the true labels generally decreases moving from less to more domain shift within a landscape, consistent with expectation, with the exception of AAV/Random vs. Designed models performing better than AAV/7 vs. Rest models (Fig 4a). Most methods exhibit similar performance in Spearman rank correlations of predictions to targets ($\rho$) within the same task. For many tasks, GP and BRR models perform as well or better than CNN models. Performance on Spearman rank correlations of uncertainties to prediction residuals ($\rho_{unc}$) is generally much worse than that on $\rho$, with some results showing negative correlation (Fig 4b). MVE and evidential uncertainty methods are most performant in $\rho_{unc}$ for most cases of low to moderate domain shift. Most methods have $\rho_{unc}$ near zero for the most challenging splits. Despite the relatively good performance of MVE on tasks with low to moderate domain shift, it performs poorly in cases of high domain shift, which is consistent with its intended use as an estimator of aleatoric (data-dependent) uncertainty.

We find that the models trained on ESM embeddings outperform those trained on one-hot encodings in 21 out of 51 cases for rank correlation of test set predictions, and 29 out of 51 cases for rank correlation of test set uncertainties. The relative performance of the two representations on prediction and uncertainty rank correlation is shown in Fig D in S1 Appendix. In terms of predictions, ESM embeddings often yield substantially better performance for tasks with high domain shift (e.g. GB1/1 vs. Rest and Meltome/Random), while OHE performs slightly better on tasks with lower domain shift (e.g. AAV/Random and GB1/3 vs. Rest). The relative uncertainty rank correlation performance, on the other hand, does not have a clear relationship to domain shift.

Since there is no single best UQ method across datasets, splits, and metrics, it is prudent for practitioners to quantify the performance of uncertainty estimates on each new task and to prioritize metrics according to the situation (e.g. prioritize high coverage over low width in high-risk or safety-critical situations).

## Active learning

In protein engineering, the purpose of uncertainty estimation is typically to intelligently prioritize sample acquisition to facilitate downstream experimentation. One such use case is in active learning, where uncertainty estimates are used to inform sampling with the goal of improving model predictions overall (i.e., to achieve an accurate model with less training data; Fig 5a). Having assessed the calibration and accuracy of the panel of UQ methods above, we next evaluated whether uncertainty-based active learning could make the learning process more sample-efficient. Across all datasets and splits using the pretrained ESM embeddings, data acquisition was simulated as iterative selection from the data library according to a given sampling strategy (acquisition function; see Methods for details). The results are summarized in Fig 5 for Spearman rank correlation ($\rho$) on three methods and one split per landscape, and additional results are shown in the Figs H to BH in S1 Appendix for other metrics, uncertainty methods, and splits. Across most models, the performance difference between the start of active learning (10% of training data) and end of active learning (100% of training data) is

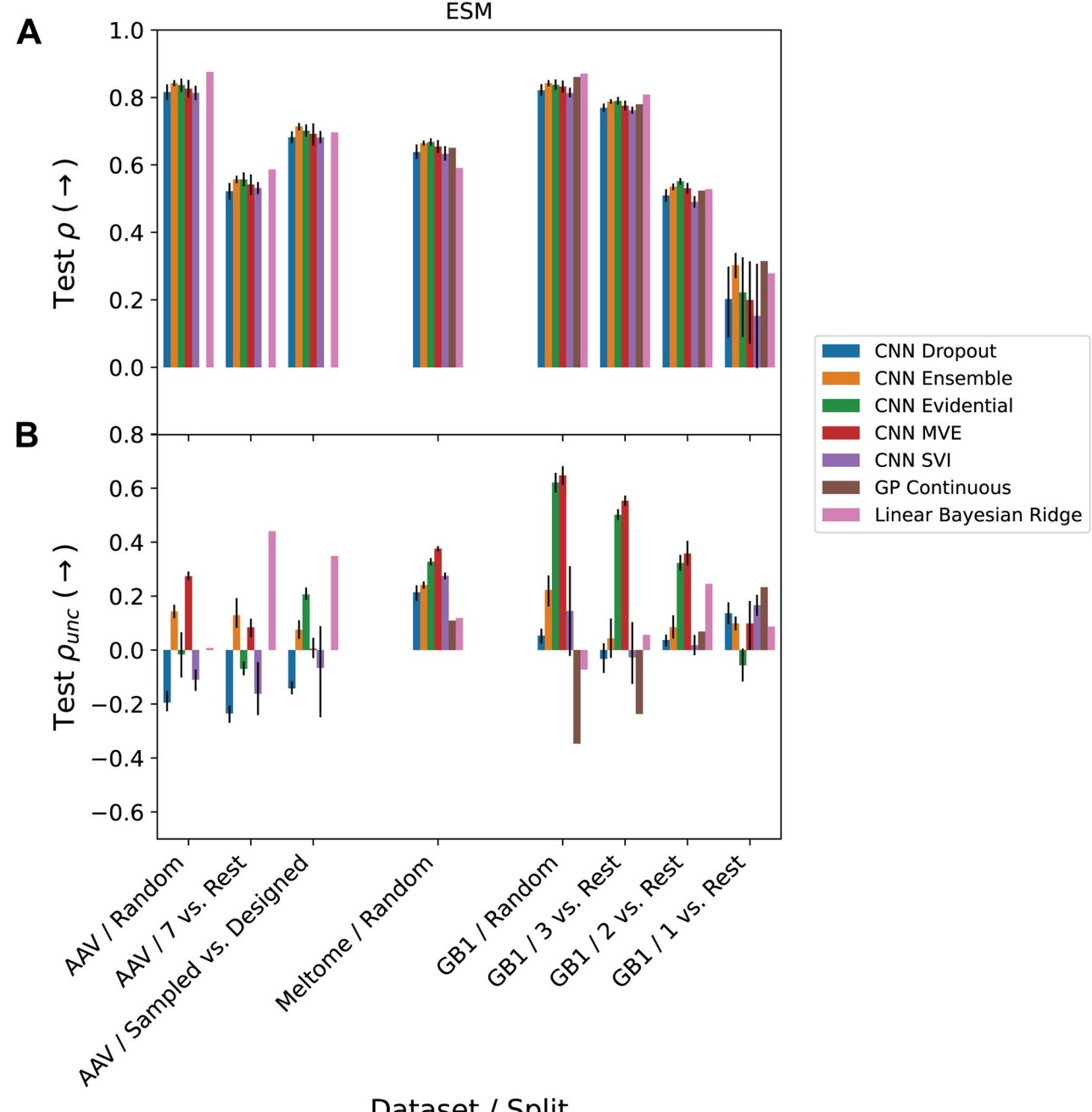

**Fig 4. Spearman rank correlations.** Of (A) predictions ($\rho$) and (B) uncertainties ($\rho_{unc}$) vs. extrapolation. Within each landscape (AAV, Meltome, and GB1), splits are ordered by the amount of domain shift between train and test sets, with the lowest domain shift on the left and the highest domain shift on the right. Error bars on the CNN results represent the 95% confidence interval calculated from 5 different random seeds for initialization of the CNN parameters and batching / stochastic gradient descent. Fig C in S1 Appendix shows the corresponding results for the OHE representation. See the Uncertainty Methods section for an explanation of several points for which experiments were not feasible.

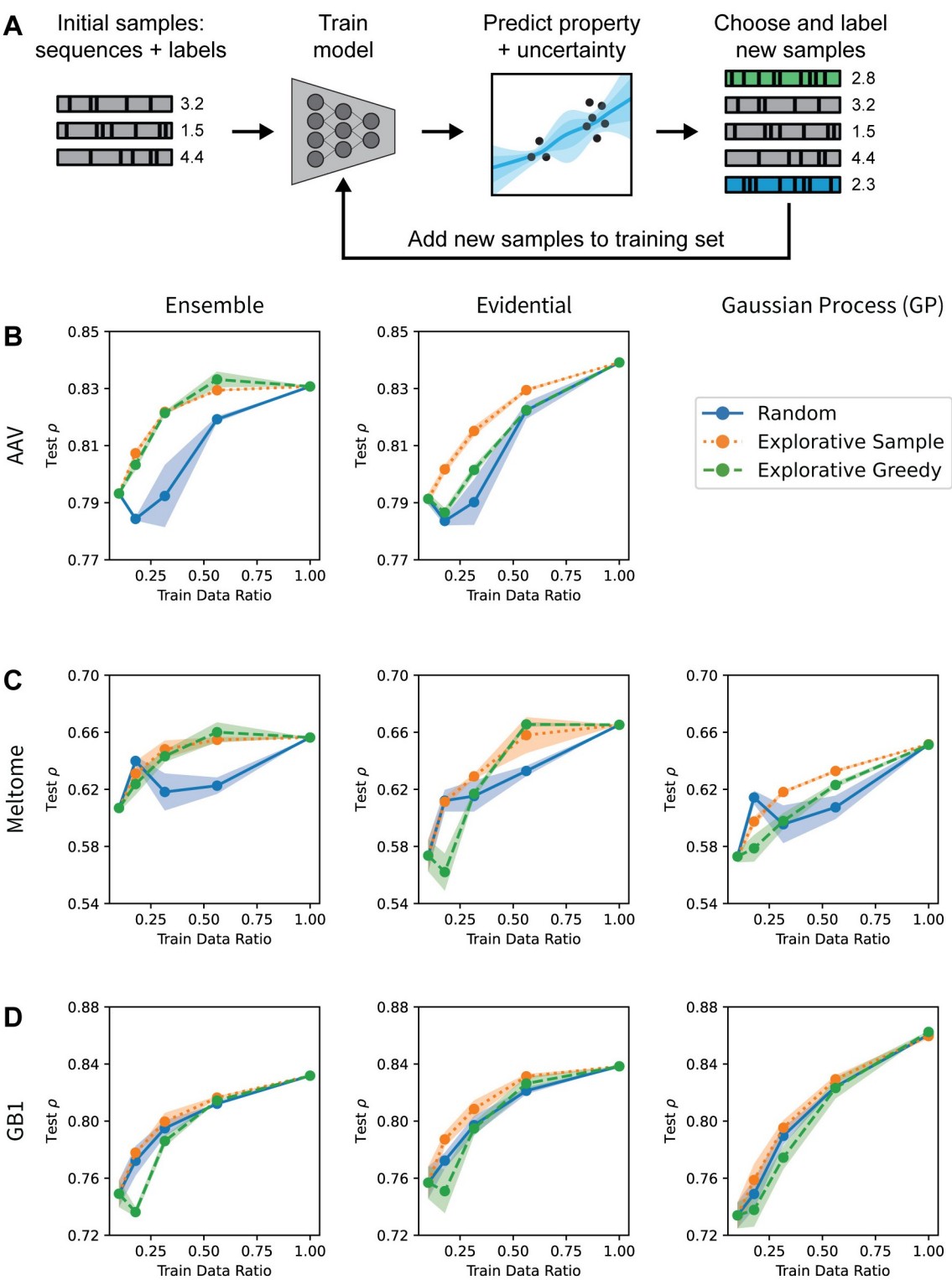

**Fig 5. Active learning.** (A) Schematic of active learning approach. A model is trained on an initial dataset, and is then retrained in each iteration by adding more points to the training set based on some selection criteria. (B-D) Uncertainty-guided active learning in protein sequence-function prediction. Spearman rank correlation of predictions ($\rho$) for the CNN ensemble, CNN evidential, and GP methods evaluated on the AAV/Random (B), Meltome/Random (C), and GB1/Random (D) splits. The "random" strategy acquired sequences with all unseen points having equal probabilities, the "explorative sample" strategy acquired sequences with random sampling weighted by uncertainty, and the "explorative greedy" strategy acquired the previously unseen sequences with the highest uncertainty. See the Uncertainty Methods section for an explanation of why GP experiments for the AAV landscape were not feasible.

relatively small, and many models begin to plateau in performance before reaching 100% of training data. In addition to the active learning experiments run with 10% of the training data in the initial sample, we also show results starting with 1% and 5% of the training data in Figs F and G in S1 Appendix, respectively. These results showed worse initial performance given the smaller initial training data, but otherwise similar trends to the trials starting at 10%.

The "explorative greedy" and "explorative sample" acquisition functions (which sample based on uncertainty alone or sample randomly weighted by uncertainty, respectively) sometimes outperform random sampling, but this is not true across all methods and landscapes (Fig 5b–5d). In some cases, the performance of the uncertainty-based sampling strategies also varies depending on the fraction of the total training data available to the model. For example, for the Meltome/Random split and CNN evidential model (Fig 5c), explorative greedy sampling results in a decrease in model performance after the first round of active learning while the explorative sample strategy increases performance. By the fourth round of active learning for this task, the two explorative strategies outperform random sampling. This indicates that in the early stages of active learning when a model's uncertainty estimates are poorly calibrated, it may be advantageous to sample with at least some randomness included in an uncertainty-based acquisition function. We also analyzed how the mean test set uncertainty changed as more data was acquired during active learning. Fig E in S1 Appendix illustrates that in some cases, the mean test set uncertainty decreased with increasing training data, while in other cases, it increased. While one might expect adding more data from uncertain sequences would always cause a decrease in the mean uncertainty, this assumes that (1) the added training data comes from the same distribution as the test data, and (2) the uncertainty estimates are well-calibrated at the beginning and remain so after retraining with more data. Overall, the results indicate that uncertainty-informed active learning can outperform random sampling and thus lead to more accurate machine learning models with fewer training points needing to be measured (Fig 5b–5d).

## Bayesian optimization

Uncertainty estimates can also be leveraged to identify top-performing sequences with as few samples as possible. The true objective can be approximated with a surrogate model, and we can use the predictions of this model as well as the uncertainties in these model predictions to guide a search toward higher or lower values of the true objective. This approach, referred to as Bayesian optimization, can also be represented by Fig 5a. Bayesian optimization methods use an acquisition function computed from the predicted mean and uncertainty values to trade off exploration and exploitation when choosing new sequences to sample, in contrast to the acquisition functions used in active learning that maximize exploration.

We compared two popular acquisition functions (upper-confidence bound (UCB) and Thompson sampling (TS)) against random and greedy baselines (see Methods for details). UCB and TS are intended to accelerate identification of top-performing instances over random and greedy baselines by taking into account both the predicted objective and the uncertainty in that prediction. Fig 6 shows the % of top-100 scores found versus fraction of training data seen for three UQ methods and one split per landscape. Across these cases, the uncertainty-based methods almost always perform better than the random baseline but never outperform greedily sampling the sequences with the highest predicted values. In most cases, UCB sampling performs about the same as greedy sampling, with the notable exception of evidential uncertainty on the AAV sampled dataset (Fig 6a), for which it performed worse than random sampling. The performance of TS (a probabalistic method) was typically intermediate between greedy/UCB (deterministic methods) and random sampling. Overall, the

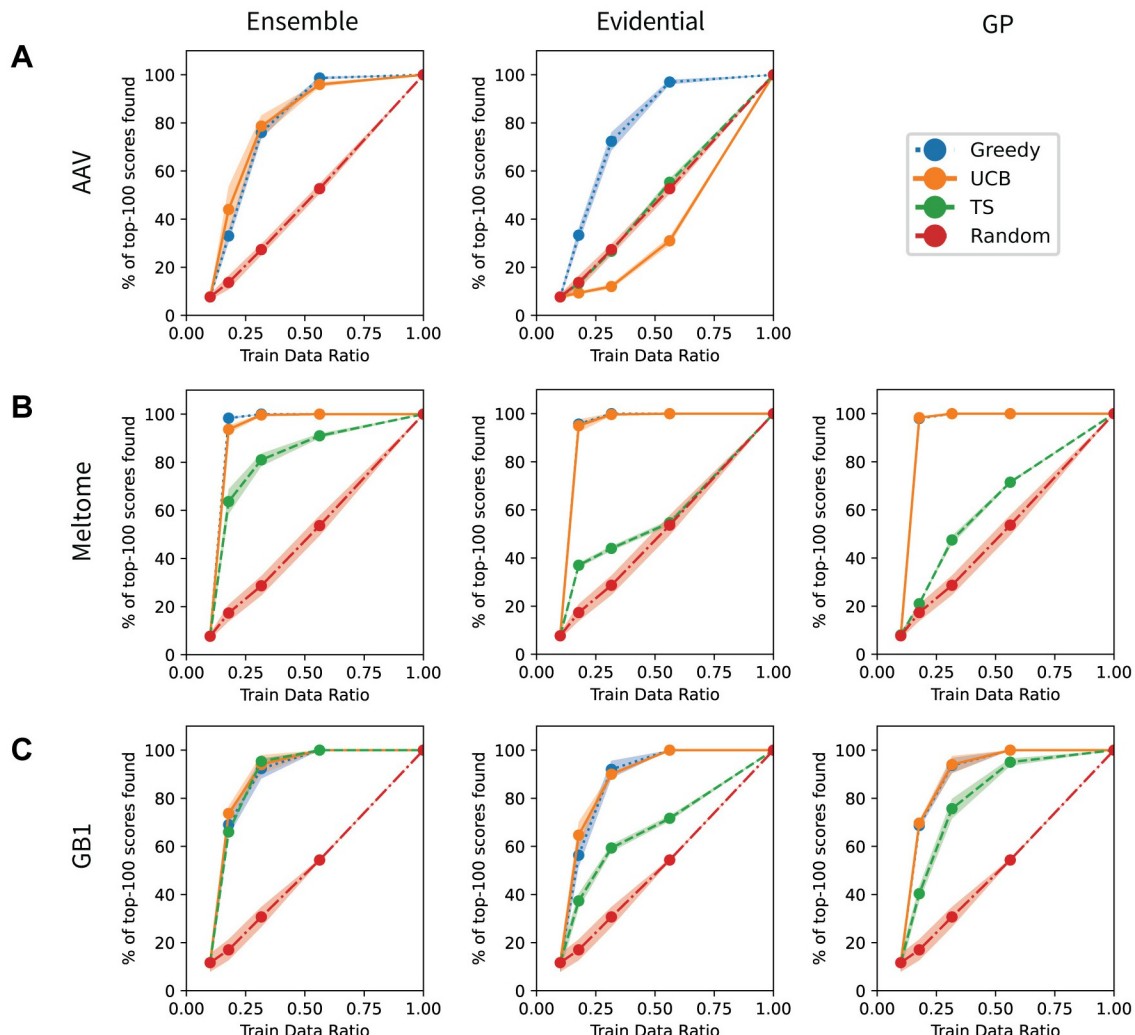

**Fig 6. Bayesian optimization.** (A-C) Bayesian optimization in protein sequence-function prediction. % of top-100 scores in training set found for the CNN ensemble, CNN evidential, and GP methods evaluated on the AAV/Random (A), Meltome/Random (B), and GB1/Random (C) splits. The "greedy" strategy acquired sequences with the best predicted property values. The "UCB" and "TS" strategies acquired sequences based on the upper confidence bound (UCB) and Thompson sampling (TS) approaches, respectively. The "random" strategy acquired sequences with all unseen points having equal probabilities. See the Uncertainty Methods section for an explanation of why GP experiments for the AAV landscape were not feasible. Note that in several plots, including the Gaussian process plots for Meltome and GB1 and the evidential plot for Meltome, the "greedy" strategy performance is nearly identical to and is covered up by the "UCB" strategy.

uncertainty-based methods do not outperform greedy baselines, suggesting that better UQ methods are needed for protein engineering or that the landscapes studied here are simple enough that they can be optimized by pure exploitation.

## Conclusions

Calibrated uncertainty estimations for ML predictions of biomolecular properties are necessary for effective model improvement using active learning or property optimization using Bayesian methods. In this work, we benchmarked a panel of uncertainty quantification (UQ) methods on protein datasets, including on train-test splits that are representative of real-world

data collection practices. After evaluating each method based on accuracy, calibration, coverage, width, rank correlation, and performance in active learning and Bayesian optimization, we find that there is no method that performs consistently well across all metrics or all landscapes and splits.

We also examined how models trained using one-hot-encoding representations of sequences compare to those trained on more informative and generalizable representations such as embeddings from a pretrained ESM language model. This comparison illustrated that while the pretrained embeddings do improve model accuracy and uncertainty correlation/calibration in some cases, particularly on splits with higher domain shifts, this is not universally true and in some cases makes performance worse.

While the UQ evaluation metrics used in this work provide valuable information, they are ultimately only a proxy for expected performance in Bayesian optimization and active learning. We found that UQ evaluation metrics are not well-correlated with gains in accuracy from one active learning iteration to another on these datasets. This suggests that future work in UQ should include retrospective Bayesian optimization and/or active learning studies rather than relying on UQ evaluation metrics alone. Our retrospective active learning studies using hold-outs of the training sets demonstrate that many of the uncertainty methods outperform random sampling baselines. In some of our experiments, we observe that the uncertainty-based sampling strategies perform worse than random sampling during the earliest stages of active learning, then perform better as a model's accuracy and quality of uncertainty estimates improve in later stages. Our Bayesian optimization experiments demonstrate that while uncertainty-based methods typically perform better than a random approach, including uncertainty in the acquisition function does not necessarily confer a benefit over a greedy approach that considers only property predictions. While previous work has successfully used BO to optimize proteins [24–27], it is not clear that uncertainty helped these campaigns because they do not compare directly to greedy sampling. Taken together, these results indicate that there is a need for further development of UQ methods and/or sampling strategies to improve protein engineering performance in AL and BO settings.

Future work in this area could expand on methods (e.g. Bayesian neural networks [28] and conformal prediction [29, 30]), metrics (e.g. sharpness [5], dispersion [31], and tightness [32]), and representations (e.g. ESM-2 [33] or using an attention layer rather than mean aggregation on our ESM-1b embeddings). In addition to further study of existing methods, future work should focus on designing novel UQ methods that give a clear performance benefit in AL and BO for protein engineering. Future work could also examine other sampling regimes for AL and BO, such as training an initial model on random data and sampling from designed sequences. While this work considered uncertainty predictions as directly output by the models, further study is needed to understand the effects of post-hoc calibration methods (e.g. scalar recalibration [31] or CRUDE [34]). Future work should consider additional active learning and Bayesian optimization strategies, such as those that consider batch diversity in the acquisition function [35], and methods that consider the desired domain shift. Ultimately, this work contributes to a more thorough understanding of the performance and utility of UQ for sequence-function models and provides a foundation for future work to enable more effective protein engineering.

## Methods

### Regression tasks

All tasks studied in this work are regression problems, in which we attempt to fit a model to a dataset with $\mathcal{D}$ data points $(x_i, y_i)$. $x_i$ is a protein sequence representation (either a one-hot

encoding or an embedding vector from an ESM language model), and $y_i \in \mathbb{R}$ is a scalar-valued target property from the protein landscapes described in the Datasets section.

## Datasets

The landscapes and splits in this work are taken from the FLIP benchmark [3]. GB1 is a landscape commonly used for investigating epistasis (interactions between mutations) using the binding domain of protein G, an immunoglobulin binding protein in Streptococcal bacteria. These splits are designed primarily to test generalization from few- to many-mutation sequences. The AAV landscape is based on data collected for the Adeno-associated virus capsid protein, which help the virus integrate a DNA payload into a target cell. The mutations in this landscape are restricted to a subset of positions within a much longer sequence. The Meltome landscape includes data from proteins across 13 different species for a non-protein-specific property (thermostability), so it includes both local and global variations. The total number of data points in the GB1, AAV, and Meltome sets are 8,733, 284,009, and 27,951, respectively. In the AAV set, 82,583 are sampled (mutations) and 201,426 are designed. For AAV, only the 82,583 sampled sequences are used for the Random and 7 vs. Rest tasks, while all 284,009 are used for the Sampled vs. Designed task.

 The names of several of the tasks were changed slightly from the original FLIP nomenclature for clarity: GB1/Random was originally called GB1/Sampled, AAV/Random was originally called AAV/Sampled, AAV/7 vs. Rest was originally called AAV/7 vs. Many, AAV/Sampled vs. Designed was originally called AAV/Mut-Des, and Meltome/Random was originally called Meltome/Mixed.

## ESM embeddings

We used the pretrained, 650M-parameter ESM-1b model (`esm1b_t33_650M_UR50S`) from [15] to generate embeddings of the protein sequences in this study and to compare these embeddings to one-hot encoding representations. Sequence embeddings from the final representation layer (layer 33) were mean pooled per amino acid over the length of each protein sequence, which resulted in a fixed embedding size of 1280 for each sequence. In other words, the output of the ESM-1b model is a tensor of size $L \times 1280$, and we averaged over each sequence to obtain a representation vector of size 1280 for each sample.

## Base CNN model architectures

The base architecture of all CNN models in this work was taken from the CNNs in the FLIP benchmark [3], which took the architecture from previous work [36]. For the one-hot encoding inputs (with a vocabulary of 22 tokens), this was comprised of a convolution with 1024 output channels and kernel width 5, a ReLU non-linear activation function, a linear mapping to 2048 dimensions, a max pool over the sequence, and a linear mapping to 1 dimension. For ESM embedding inputs (of size 1280), the architecture was the same except with 1280 input channels rather than 1024, and a linear mapping to 2560 dimensions rather than 2048.

## CNN Model training procedures

To train our CNN models, we used a batch size of 256 (GB1, AAV) or 30 (Meltome). Adam [37] was used for optimization with the following learning rates: 0.001 for the convolution weights, 0.00005 for the first linear mapping, and 0.000005 for the second linear mapping. Weight decay was set to 0.05 for both the first and second linear mappings. CNNs were trained with early stopping using a patience of 20 epochs. Each model was trained on an NVIDIA

Volta V100 GPU. Reported metrics in Figs 2–4 are the average of training 5 models per split with different seeds for initialization of the CNN parameters and batching / stochastic gradient descent. Code, data, and instructions needed to reproduce results can be found at https://github.com/microsoft/protein-uq.

## Uncertainty methods

For all models and landscapes, the sequences were featurized using either one-hot encodings or embeddings from a pretrained language model (see the ESM Embeddings section).

We used the `scikit-learn` [38] implementation of Bayesian ridge regression (BRR) with default hyperparameters. BRR for one-hot encodings of the Meltome/Random split was not feasible because the required work array was too large to perform the computation with standard 32-bit LAPACK in `scipy`.

For Gaussian processes (GPs), we used the GPyTorch [39] implementation with the constant mean module, scaled rational quadratic (RQ) kernel covariance module, and Gaussian likelihood. Some GP models (for AAV one-hot encodings and ESM embeddings, and Meltome one-hot encodings) were not feasible to train due to GPU-memory requirements for exact GP models, so these are omitted from the results.

For our uncertainty methods that rely on sampling (dropout, ensemble, and SVI), the final model prediction is defined as the mean of the set of inference samples, and the uncertainty is the standard deviation of these samples. In other words, for a set of predictions $\mathcal{E} = \{G_1(x), G_2(x), \ldots, G_n(x)\}$ (each coming from an individual model $G_i$), the final prediction is defined as

$$\hat{G}(x) = \sum_{G \in \mathcal{E}} \frac{G(x)}{n} \tag{1}$$

and the uncertainty $U(x)$ is defined as

$$U(x) = \sqrt{\sum_{G \in \mathcal{E}} \frac{(\hat{G}(x) - G(x))^2}{n}} \tag{2}$$

The uncertainty is sometimes defined as the variance $U^2$, but using the standard deviation puts the uncertainty in the same units as the predictions.

For dropout uncertainty [19], a single model $G$ was trained normally. At inference time, we applied $n = 10$ random dropout masks with dropout probability $p$ to obtain the set of predictions $\mathcal{E}$ for each input $x_i$. We tested dropout rates of $p \in \{0.1, 0.2, 0.3, 0.4, 0.5\}$ and reported the model with the lowest miscalibration area.

Similarly for last-layer stochastic variational inference (SVI) [23], we obtained $\mathcal{E}$ using $n = 10$ samples from a set of models where each $G_i$ has the weight and bias terms of its last layer themselves sampled from a distribution $q(\theta)$ that has been trained to approximate the true posterior $p(\theta|\mathcal{D})$.

Traditional model ensembling calculated $\mathcal{E}$ using $n = 5$ models trained using different random seeds for initialization of the CNN parameters and batching / stochastic gradient descent. The computational cost of this approach is 5 times that of a standard CNN model since the cost scales linearly with the size of the ensemble.

In mean-variance estimation (MVE) models, we adapt the base CNN architecture to produce 2 outputs ($\theta = \{\mu, \sigma^2\}$) for each data point ($x_i, y_i$) in the last layer rather than 1, and we

train using the negative log-likelihood loss:

$$\mathcal{L}(\theta) = \frac{1}{N}\sum_{i=1}^{N}\frac{(y_i - \mu(x_i))^2}{2\sigma^2(x_i)} + \frac{1}{2}\log(2\pi\sigma^2(x_i)) \tag{3}$$

In practice, the variance ($\sigma^2$) is clamped to a minimum value of $10^{-6}$ to prevent division by 0.

Evidential deep learning modifies the loss function of the traditional CNN to jointly maximize the model's fit to data while also minimizing its evidence on errors (increasing uncertainty on unreliable predictions) [21]:

$$\mathcal{L}(x) = \mathcal{L}^{NLL}(x) + \lambda\mathcal{L}^{R}(x) \tag{4}$$

where $\mathcal{L}^{NLL}(x)$ is the negative log-likelihood loss defined above, $\mathcal{L}^{R}(x)$ is the evidence regularizer as defined in Amini et al. [21], and $\lambda$ controls the trade-off between these two terms. In this study, we use $\lambda = 1$ for all evidential models. In these models, the last layer of the model produces 4 outputs $\mathbf{m} = \{\gamma, \nu, \alpha, \beta\}$ that parameterize the Normal-Inverse-Gamma distribution. This distribution assumes that targets $y_i$ are drawn i.i.d. from a Gaussian distribution with unknown mean and variance $\theta = \{\mu, \sigma^2\}$, where the mean is drawn from a Gaussian and the variance is drawn from an Inverse-Gamma distribution. The output of the evidential model can be divided into the prediction and the epistemic (model) and aleatoric (data) uncertainty components following the analysis of Amini et al. [21]:

$$\underbrace{\mathbb{E}[\mu] = \gamma}_{\text{prediction}}, \quad \underbrace{\mathbb{E}[\sigma^2] = \frac{\beta}{\alpha - 1}}_{\text{aleatoric}}, \quad \underbrace{\mathrm{Var}[\mu] = \frac{\beta}{\nu(\alpha - 1)}}_{\text{epistemic}} \tag{5}$$

We report the sum of the aleatoric and epistemic uncertainties as the total uncertainty.

## Evaluation metrics

To give a comprehensive report of model accuracy, we computed the following metrics on the test sets: root mean square error (RMSE), mean absolute error (MAE), coefficient of determination ($R^2$), and Spearman rank correlation ($\rho$). RMSE is more sensitive to outliers than MAE, so while both are informative independently, the combination of the two gives additional information about the distribution of errors. $R^2$ and $\rho$ are both unitless and are thus more easily interpreted and compared across datasets.

We evaluated the quality of the uncertainty estimates using four metrics. First, $\rho_{unc}$ is the Spearman rank correlation between uncertainty and absolute prediction error. This metric may be particularly relevant in an active learning context, where one wants to acquire labels for the most uncertain points hoping that these are also the highest-error points. This application does not require the uncertainties to be well-calibrated.

Second, the miscalibration area (also called the area under the calibration error curve or AUCE) quantifies the absolute difference between the calibration plot and perfect calibration in a single number [40]. Good calibration may be more important in safety-critical applications.

Following Kompa et al. [41], we measured the coverage as the percentage of true values that fall within the 95% confidence interval ($\pm 2\sigma$) of each prediction. This is a indication of the reliability of the uncertainty estimates. A model with high coverage is appropriately cautious in its predictions, which may be most important in applications where safety is a major consideration.

Kompa et al. [41] also define another metric, the width, as the size of the 95% confidence region ($4\sigma$). We normalized this width relative to the range ($R$) of the training set as $4\sigma/R$ to

make these values unitless and thus more interpretable across datasets. The width is a measure of precision in the uncertainty. In practical applications, narrower intervals (lower width) can help in making more precise and cost-effective decisions. Ideal uncertainties have high coverage and low width, but in some cases, there may be trade-offs between the two. For example, wider widths can help to detect distribution shift, but these wider intervals may not be reliable if coverage is low. The coverage and width metrics may also be more easily interpretable than other calibration metrics [41].

### Active learning

Each active learning run began with a random sample of 1%, 5%, or 10% of the full training data, which was taken from the random splits of the three landscapes. We evaluated several alternatives for adding to this initial dataset using different sampling strategies (acquisition functions): explorative greedy, explorative sample, and random. "Explorative greedy" sampled the sequences with the highest uncertainty; "explorative sample" sampled the data according to the probability of sampling a sequence equal to the ratio of its uncertainty to the sum of all uncertainties in the dataset (i.e. random sampling weighted by uncertainty); and "random" sampled the data uniformly from all unobserved sequences. We employed these sampling strategies 5 times in each active learning run, with the 5 training set sizes equally spaced on a log scale. We repeated this process using 3 folds (different random seeds for sampling initial dataset and "explorative sample" probabilities) and calculated the mean and standard deviation across these folds.

### Bayesian optimization

Similarly to active learning, each Bayesian optimization run began with a random sample of 10% of the full training data, which was taken from the random splits of the three landscapes. We used the following acquisition functions: greedy, upper-confidence bound (UCB) [42], Thompson sampling (TS) [43], and random. In greedy sampling, the sequence with the best predicted value was selected. The UCB strategy added the predicted uncertainties to the predicted values and selected the sequence with the largest sum. For TS, we added each predicted value to a number sampled randomly from a Gaussian distribution with a mean of 0 and a standard deviation of the corresponding predicted uncertainty, and again selected the largest sum. The "random" strategy used in Bayesian optimization was the same as that used in active learning (new points were sampled with uniform probability). As with active learning, we used these strategies 5 times in each run, with the 5 training set sizes equally spaced on a log scale. We report the mean and standard deviation across 3 folds (different random seeds for sampling the initial dataset).

### Supporting information

**S1 Appendix. Supporting information.** Code availability, OHE results, OHE vs. ESM comparison, additional prediction and uncertainty evaluation metrics, and additional active learning results.
(PDF)

### Acknowledgments

The authors thank the MIT Lincoln Laboratory Supercloud cluster [44] at the Massachusetts Green High Performance Computing Center (MGHPCC) for providing high-performance computing resources to train our machine learning models.

## Author Contributions

**Conceptualization:** Ava P. Amini, Kevin K. Yang.

**Formal analysis:** Kevin P. Greenman.

**Investigation:** Kevin P. Greenman.

**Methodology:** Kevin P. Greenman.

**Software:** Kevin P. Greenman.

**Supervision:** Ava P. Amini, Kevin K. Yang.

**Validation:** Kevin P. Greenman.

**Visualization:** Kevin P. Greenman.

**Writing – original draft:** Kevin P. Greenman.

**Writing – review & editing:** Ava P. Amini, Kevin K. Yang.

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
