## [Decision Letter · Decision Letter 0]

11 Feb 2024

Dear Yang,

Thank you very much for submitting your manuscript "Benchmarking uncertainty quantification for protein engineering" for consideration at PLOS Computational Biology.

As with all papers reviewed by the journal, your manuscript was reviewed by members of the editorial board and by several independent reviewers. In light of the reviews (below this email), we would like to invite the resubmission of a significantly-revised version that takes into account the reviewers' comments.

We cannot make any decision about publication until we have seen the revised manuscript and your response to the reviewers' comments. Your revised manuscript is also likely to be sent to reviewers for further evaluation.

Thank you again for your submission. We hope that our editorial process has been constructive so far, we apologize for the long time it took, and we welcome your feedback at any time. Please don't hesitate to contact us if you have any questions or comments.

Sincerely,

Rachel Kolodny

Academic Editor

PLOS Computational Biology

Nir Ben-Tal

Section Editor

PLOS Computational Biology

Reviewer's Responses to Questions

**Comments to the Authors:**

Reviewer #1: This paper is about quantifying the ability of different protein sequence-based models to accurately predict uncertainty in different protein design data regimes and modalities. The authors tried difference sequence representations, model architectures and uncertainty quantification methods across FLIP benchmark tasks. Although the amount of experiments involved in this work is impressive, the paper is unfortunately written in a way that mostly lists those dense results, hence making it difficult to get any insight or grasp the relationship between them. Moreover, some of the paper's claims could benefit from better statistical analysis support.

**Major comments**

The paper is results-dense and would benefit from some curation of metrics and/or models that would make it easier to ingest. For example, I suspect that RMSE and correlation are highly correlated as well as calibration plot and uncertainty correlation, but they both have their own distinct figures. Some of these results could be moved to supplementary.

The figures are hard to read or interpret since they tend to have too much information and don’t use the space efficiently by repeating the same axis and legend that end up using all the space. For example, the bar plots in Figure 4 are hard to read, especially the B panel for models that have close to zero correlation that seems to be missing.

Can the authors comment on the apparent strong relationship found between the coverage and width metrics for the same task across different models? It almost points toward models not better quantifying uncertainty per point, but just being more confident in general. It almost feels that a simple re-scaling of the uncertainties on a calibration set from training sets would make those trends and claims vanish.

Claims about models being better than others on task are hard to support without any statistical testing. The fact that using different starting seeds for training the CNN model gives similar performance across Figure 4 almost indicates that some of the trends found in previous figures could also vanish. Also, the active learning experiments with the random acquisition function seem to indicate that a simple re-sampling of the training data could be used to better estimate confidence intervals on metrics and support claims about model performance versus each other. Can the authors re-sample their training data to generate different training data and get a better estimation of their metrics? Can they also choose at least 3-4 different starting points in BO and active learning to not let the starting 10% dictate the conclusion?

Can the authors comment on the expected relationship between the different metrics? It is not clear why the miscalibration curve should be plotted against the RMSE.

**Minor comments**

The authors do cite their uncertainty metrics, but given how central they are to the paper, can they describe them more?

I might have missed these details, but what are the data used for active learning and BO. Is it the designed AAV or just the random? Would it make more sense to start from sampled and then sample from designed? (or maybe it is the case)

In the active learning experiments, are the models getting more confident as they get more details?

A summary table of the different rankings of selected models across tasks would be helpful since the paper has so many results that it is hard to keep track of performance across different regimes and modalities. Ideally, these rankings would be derived from robust statistical testing.

Some curves are not visible in Figure 6 since they are hidden by others.

Reviewer #2: This paper considers the problem of uncertainty prediction for the problem of protein engineering. The paper reports extensive experimentation, with the goal to evaluate different types of predictors, and specifically evaluate their uncertainty predictions. The experiment is conducted in three different types of tasks, with varying degree of domain shift between the training data and test data. The uncertainty prediction is evaluated using different metrics, and through two downstream applications, namely, active learning and Bayesian optimization. The results of the experiments show that:

1. There is no single method that performs better across all tasks.

2. Current uncertainty predictors are not always useful for downstream tasks like active learning (where they are useful in some cases), and Bayesian optimization (where a greedy, uncertainty-agnostic approach performs better).

3. Representing proteins using the ESM language model embeddings rather than one-hot encoding, improves results in some cases but not all.

The main conclusion is that uncertainty predictors cannot be assumed to work out of the box for protein engineering tasks and need to be further developed, and/or carefully evaluated per task.

Strengths:

1. The paper performs an extensive and thorough evaluation of the methods under different varying conditions, and gives both a detailed description of each setup and the big picture of the status of current methods.

2. The conclusion is a useful and practical contribution to the community that often considers relying on such predictions of uncertainty.

Weaknesses:

1. The paper does not present a novel model or evaluation methodology, and is a relatively straightforward implementation of various experiments. In my view this should not prevent the paper from being accepted as the experimentation is thorough and therefore valuable to the community.

2. The results do not show a clear winning method that can directly inform practitioners on ways to improve their research on downstream tasks. However, like I mentioned above, there is value in empirically demonstrating the limitations of current models, which should serve as a warning for practitioners of downstream tasks, and an invitation to researchers to perform more research on uncertainty prediction.

3. There are a few issues that were not clear to me. See questions below. I believe that fixing those issues, would make the paper publishable.

Questions:

1. The representation and CNN architecture is not clear to me. What dimension exactly is being averaged in the ESM embeddings? At the end, what is the dimension of the protein representation both for ESM and one-hot encoding? Is it constant or varying with sequence length? Why do you need a CNN to process this representation, as opposed to a fully connected MLP? Is there some local invariance or smoothness property that should be captured by convolutions? If so over what dimension?

2. It is not clear from the description in section 4.6 that all methods use the same number of samples in evaluation. If this is the case it should be stated, and if not, it should be discussed and justified.

3. Some methods were not fully described, for example it was mentioned that the evidential CNN uses a loss termed L^R, without elaborating.

3. Section 4.7 is confusing in listing the different metrics. First it describes MAE and R^2 metrics which I failed to see in any of the epxeriment results. Second, it states there are four uncertainty metrics but I could only count three (ro_unc, coverage, AUCE). Third, the description of the coverage states that the range is 4\\sigma/R rather than 4\\sigma, however in figure 3 these are shown on different axes (coverage vs. width/R) - I’m not sure what’s going on there.

Some general remarks:

1. The fact that the linear model and GP are performing better than the CNN suggest that maybe there is not enough training data for a deep learning method to work. Are there larger datasets that can be tested?

2. I’m not sure there is a way to do this better than the thorough experiments setup already presented, but in the evaluation through downstream tasks (active learning and Bayesian optimization), there is still some conflation between the quality of the prediction and the quality of the uncertainty predictions. It might be beneficial to try to compare a predictor where the uncertainty is computed exactly from ground truth data. This can serve as some kind of upper bound on the performance.

**Have the authors made all data and (if applicable) computational code underlying the findings in their manuscript fully available?**

Reviewer #1: Yes

Reviewer #2: Yes

PLOS authors have the option to publish the peer review history of their article (what does this mean?). If published, this will include your full peer review and any attached files.

Reviewer #1: No

Reviewer #2: No
---

## [Decision Letter · Decision Letter 1]

29 Aug 2024

Dear Yang,

Thank you very much for submitting your manuscript "Benchmarking uncertainty quantification for protein engineering" for consideration at PLOS Computational Biology. As with all papers reviewed by the journal, your manuscript was reviewed by members of the editorial board and by several independent reviewers. The reviewers appreciated the attention to an important topic. Based on the reviews, we would like to accept this manuscript for publication, but please modify the small changes the reviewer requested. 

Sincerely,

Rachel Kolodny

Academic Editor

PLOS Computational Biology

Nir Ben-Tal

Section Editor

PLOS Computational Biology

Reviewer's Responses to Questions

**Comments to the Authors:**

Reviewer #1: The author has addressed most of my comments.

Regarding the statistical evidence discussion, I do understand that creating new splits is its own endeavor, and this paper is not about FLIP but about uncertainty quantification. However, since the author already did the hard work of training 5 models per split with different seeds, can they also report the standard deviation for all metrics in the supplementary table. This could help the reader have some context on how different models perform relative to each other in a given context

**Have the authors made all data and (if applicable) computational code underlying the findings in their manuscript fully available?**

Reviewer #1: Yes

PLOS authors have the option to publish the peer review history of their article (what does this mean?). If published, this will include your full peer review and any attached files.

Reviewer #1: No

Figure Files:

Data Requirements:

Reproducibility:

References:

---

## [Editor Report · Decision Letter 2]

14 Nov 2024

Dear Yang,

We are pleased to inform you that your manuscript 'Benchmarking uncertainty quantification for protein engineering' has been provisionally accepted for publication in PLOS Computational Biology.

Best regards,

Rachel Kolodny

Academic Editor

PLOS Computational Biology

Nir Ben-Tal

Section Editor

PLOS Computational Biology

Feilim Mac Gabhann

Editor-in-Chief

PLOS Computational Biology

Jason Papin

Editor-in-Chief

PLOS Computational Biology

---

## [Editor Report · Acceptance letter]

11 Dec 2024

PCOMPBIOL-D-23-01757R2 

Benchmarking uncertainty quantification for protein engineering

Dear Dr Yang,

I am pleased to inform you that your manuscript has been formally accepted for publication in PLOS Computational Biology. Your manuscript is now with our production department and you will be notified of the publication date in due course.

With kind regards,

Dorothy Lannert
